# The nhppp package for simulating non-homogeneous Poisson point processes in R

**Thomas A. Trikalinos**[1,2,3]*, **Yuliia Sereda**[1]

**1** Center for Evidence Synthesis in Health, Brown University, Providence, RI, United States of America, **2** Department of Health Services, Policy & Practice, Brown University, Providence, RI, United States of America, **3** Department of Biostatistics, Brown University, Providence, RI, United States of America

\* thomas_trikalinos@brown.edu

## Abstract

We introduce the **nhppp** package for simulating events from one dimensional non-homogeneous Poisson point processes (NHPPPs) in R fast and with a small memory footprint. We developed it to facilitate the sampling of event times in discrete event and statistical simulations. The package's functions are based on three algorithms that provably sample from a target NHPPP: the time-transformation of a homogeneous Poisson process (of intensity one) via the inverse of the integrated intensity function; the generation of a Poisson number of order statistics from a fixed density function; and the thinning of a majorizing NHPPP via an acceptance-rejection scheme. We present a study of numerical accuracy and time performance of the algorithms. We illustrate use with simple reproducible examples.

## 1 Introduction

It is often desirable to simulate series of events (stochastic point processes) so that the intensity of their occurrence varies over time. Examples include events such as the occurrence of death and occurrences of symptoms, infections, or tumors over a person's lifetime. The non-homogeneous Poisson point process (NHPPP), which generalizes the simpler homogeneous-Poisson, Weibull, and Gompertz point processes, is a widely used model for such series of events. NHPPPs can model complicated event patterns given a suitable intensity function. They are, therefore, useful in statistical and mathematical model simulation.

An NHPPP has the properties that the number of events in all non-overlapping time intervals are independent random variables and that, within each time interval, the number of events is Poisson distributed. Thus an NHPPP is a memoryless point process. A large number of phenomena may reasonably conform with these properties.

NHPPPs have been used in the simulation analysis of queues in queuing theory and operations research [1, 2]; hospital operations [3]; ambulance services [4, 5]; traffic accidents [6]; product and network reliability [7]; and the modeling of cancer [8–11], heart disease [12], and dementia [13, 14], among other applications [15]. NHPPPs are used so widely in part because their assumptions are often plausible. For example, when modeling traffic accidents along a road, it may be plausible to assume that individual accidents are independent of each other, but they happen in some locations more often because the probability of an accident depends

**Data Availability Statement:** There are no data associated with the paper. All the code to generate the exhibits is provided as supplementary information.

**Funding:** TAT and YS were funded from grant U01CA265750 from the National Cancer Institute (https://www.cancer.gov). The funder played no role in the study design, analysis, decision to publish, or the preparation of the manuscript.

**Competing interests:** The authors have declared that no competing interests exist.

on local aspects of the road, such as turns, slopes, and propensity for slippery conditions. Similarly, when modeling the impact of screening strategies on colorectal cancer outcomes at the population level, it is probably plausible to assume that, for each person, the emergence of precancerous lesions (adenomas) over a time interval is independent of whether such lesions emerged in other non-overlapping time intervals. In these examples, the intensity of event occurrence over the carrier space (the probability of a traffic accident along a road; and the probability that an adenoma will emerge in a person's colon at different ages) is captured by the NHPPP's intensity function. An NHPPP can model complicated event patterns using intensity functions that vary over the carrier space (e.g., length of road, time).

The **nhppp** package in R contains functions for the simulation of NHPPPs over a one-dimensional carrier space, which we will take to represent time [16]. Table 1 summarizes the functions implemented in **nhppp** as of version 0.1.4. You can install the development version of **nhppp** with

**Table 1. Functions in nhppp.**

| Intensity function | Function name | Function simulates... |
|---|---|---|
| Constant | `ppp_n()` | exactly $n$ events in $[a, b]$ |
| | `ppp_next_n()` | the next $n$ events in $[a, \infty)$ |
| | `ppp_sequential()`, `ppp_orderstat()` | $\geq 0$ events in $[a, b]$ |
| | `ztppp()` | $\geq 1$ events in $[a, b]$ |
| Time-varying, special cases, non-vectorized | `draw_sc_linear()` | $\geq 0$ events in $[a, b]$ from $\lambda(t) = \alpha + \beta t$ |
| | `draw_sc_loglinear()` | $\geq 0$ events in $[a, b]$ from $\lambda(t) = \exp(\alpha + \beta t)$ |
| | `draw_sc_step()` | $\geq 0$ events in $[a, b]$ from piecewise constant $\lambda(t)$ with uneven intervals |
| | `draw_sc_step_regular()` | $\geq 0$ events in $[a, b]$ from piecewise constant $\lambda(t)$ with regular intervals |
| | `ztdraw_sc_linear()` | $\geq 1$ events in $[a, b]$ from $\lambda(t) = \alpha + \beta t$ |
| | `ztdraw_sc_loglinear()` | $\geq 1$ events in $[a, b]$ from $\lambda(t) = \exp(\alpha + \beta t)$ |
| Time varying, special cases, vectorized | `vdraw_sc_step_regular()` | $\geq 0$ events in $[a, b]$ from piecewise constant $\lambda(t)$ with regular intervals |
| | `vztdraw_sc_step_regular()` | $\geq 1$ events in $[a, b]$ from piecewise constant $\lambda(t)$ with regular intervals |
| Time-varying, general case, non-vectorized | `draw()` | (wrapper function) |
| | `draw_cumulative_intensity_inversion()` | $\geq 0$ events in $[a, b]$ from $\Lambda(t), \Lambda^{-1}(t)$ |
| | `draw_cumulative_intensity_orderstats()` | $\geq 0$ events in $[a, b]$ from $\Lambda(t), \Lambda^{-1}(t)$ |
| | `draw_intensity()` | $\geq 0$ events in $[a, b]$ from $\lambda(t), \lambda_*(t)$ |
| | `draw_intensity_step()` | $\geq 0$ events in $[a, b]$ from $\lambda(t)$ and piecewise constant $\lambda_*(t)$ |
| | `ztdraw_cumulative_intensity()` | $\geq 1$ events in $[a, b]$ from $\lambda(t), \Lambda^{-1}(t)$ |
| | `ztdraw_intensity()` | $\geq 1$ events in $[a, b]$ from $\lambda(t), \lambda_*(t)$ |
| | `ztdraw_intensity_step()` | $\geq 1$ events in $[a, b]$ from $\lambda(t)$ and piecewise constant $\lambda_*(t)$ |
| Time varying, general case, vectorized | `vdraw()` | (wrapper function) |
| | `vdraw_intensity_step_regular()` | $\geq 0$ events in $[a, b]$ from $\lambda(t)$ and piecewise constant $\lambda_*(t)$ |
| | `vztdraw_intensity_step_regular()` | $\geq 1$ events in $[a, b]$ from $\lambda(t)$ and piecewise constant $\lambda_*(t)$ |
| (Helper function) | `get_step_majorizer()` | (obtains piecewise constant $\lambda_*(t)$ from $\lambda(t)$) |

The table pertains to version 0.1.4 of **nhppp**. $\lambda(t)$ is an intensity function, $\lambda_*(t)$ a majorizer function for $\lambda(t)$, $\Lambda(t)$ the integrated intensity function, and $\Lambda^{-1}(t)$ the inverse function (preimage) of $\Lambda(t)$.

```
R> # install.packages("devtools")
R> devtools::install_github("bladder-ca/nhppp")
```

or the release version with

```
R> install.packages("nhppp")
```

We review NHPPPs in Section 2 and algorithms for sampling from constant rate Poisson point processes in Section 3. We introduce the three sampling algorithms that are implemented in the package in Section 4. We discuss special functional forms for the intensity function (constant, piecewise constant, linear, and log-linear) in Section 5. We describe **nhppp** versus other R packages that can simulate from one dimensional NHPPPs in Section 6 and present a numerical study in Section 7. We summarize in Section 8.

## 2 The Poisson point process

### 2.1 Definition

The Poisson point process is a stochastic series of events on the real line. For some sequence of events, let $N(t, t + \Delta t)$ be the number of events in the interval $(t, t + \Delta t]$. If for some positive intensity $\lambda$ and, as $\Delta t \to 0$,

$$
\begin{aligned}
\Pr[N(t, t + \Delta t) = 0] &= 1 - \lambda \Delta t + o(\Delta t), \\
\Pr[N(t, t + \Delta t) = 1] &= \lambda \Delta t + o(\Delta t), \\
\Pr[N(t, t + \Delta t) > 1] &= o(\Delta t), \text{ and} \\
N(t, t + \Delta t) &\perp N(0, t),
\end{aligned}
\tag{1}
$$

then that sequence of events is a Poisson point process. In Eq (1), the third statement demands that events occur one at a time. The fourth statement implies that the process is memoryless: For any time $t_0$, the behavior of the process is independent to what happened before that time.

### 2.2 Homogeneous Poisson point process and counting process

Assume that the next event after time $t_0$ happens at time $t_0 + X$. It follows from the above definition (see [17, par. 4.1]) that, for a constant $\lambda$, $X$ is exponentially distributed

$$
X \sim \text{Exponential}(\lambda), \tag{2}
$$

and that the number of events is Poisson distributed over the compact interval $(a, b]$, i.e.,

$$
N(a, b) \sim \text{Poisson}(\lambda(b - a)). \tag{3}
$$

Eq (2) generates the homogeneous Poisson point process $Z_1 = t_0 + X_1, Z_2 = Z_1 + X_2, \ldots$, where $Z_i$ is the time of arrival of event $i$ and $X_i$ the inter-arrival times. We will use $Z_{(j)}$ to denote the event in position $j$ when events are ordered in increasing time. Eq (3) describes the corresponding (dual) counting process $N_1 = N(t_0, Z_1), N_2 = N(t_0, Z_2), \ldots$, where $N_i$ is the total number of events from time $t_0$ to time $Z_i$. The point process (the sequence $[Z_i]$ of event times) and the counting process (the sequence $[N_i]$ of cumulants) are two sides of the same coin.

Sampling from the constant rate point process in (2) is discussed in Section 3.

## 2.3 Non homogeneous Poisson point process and counting process

When the intensity function changes over time, the homogeneous Poisson point process generalizes to its non-stationary counterpart, an NHPPP, with intensity function $\lambda(t) > 0$. For details see reference [17], par 4.2]. Then, the number of events over the interval $(a, b]$ becomes

$$N(a, b) \sim \text{Poisson}(\Lambda(a, b)), \tag{4}$$

where $\Lambda(a, b) = \int_a^b \lambda(t) \; dt$ is the integrated intensity or cumulative intensity of the NHPPP. Eq (4) describes the counting process of the NHPPP, which in turn implies a stochastic point process—a distribution of events over time.

Here the simulation task is to sample event times from the point process that corresponds to intensity function $\lambda(t)$, or equivalently, to the integrated intensity function $\Lambda(t) = \int_0^t \lambda(s) \; ds$ (Section 4). (With some abuse of notation, we define $\Lambda(t) := \Lambda(0, t)$ when $a = 0$.)

**2.3.1 A note on zero intensity processes.** In (1), $\lambda$ is strictly positive but in **nhppp** we allow it to be non-negative. If $\lambda = 0$, $\Pr[N(t, t + \Delta t) = 0] = 1$ and $\Pr[N(t, t + \Delta t) \geq 1] = 0$. This means that no events occur and the stochastic point process in the interval $(t, t + \Delta t]$ is denegerate. Allowing $\lambda(t) \geq 0$ has no bearing on the results of simulations. If

$$\lambda(t) \begin{cases} > 0, & \text{for } t \in (a, b] \\ = 0, & \text{for } t \in (b, c] \\ > 0, & \text{for } t \in (c, d] \end{cases}$$

we can always ignore the middle interval in which no events happen.

## 2.4 Properties that are important for simulation

**2.4.1 Composability and decomposability of NHPPPs.** The definition (1) implies that NHPPPs are composable [17, par. 4.2]: merging two NHPPPs with intensity functions $\lambda_1(t)$, $\lambda_2(t)$ yields a new NHPPP with intensity function $\lambda(t) = \lambda_1(t) + \lambda_2(t)$. The reciprocal is also true: one can decompose an NHPPP with intensity function $\lambda(t)$ into two NHPPPs, one with intensity function $\lambda_1(t) < \lambda(t)$ and one with intensity function $\lambda_2(t) = \lambda(t) - \lambda_1(t)$. An induction argument extends the above to merging and decomposing three or more processes.

The composability and decomposability properties are important for simulation because they

- give the flexibility to simulate several parallel NHPPPs independently versus to merge them, simulate from the merged process, and then attribute the realized events to the component processes by assigning the $i$-th event to the $j$-th process with probability $\lambda_j(Z_i)/\lambda(Z_i)$, where $\lambda(t) = \sum \lambda_j(t)$.

- motivate a general sampling algorithm (Algorithm 4, "thinning" [18]) that simulates a target NHPPP with intensity $\lambda_1(t)$ by first drawing events from an easy-to-sample NHPPP with intensity $\lambda(t) > \lambda_1(t)$, and then accepts sample $i$ with probability $\lambda_1(Z_i)/\lambda(Z_i)$.

**2.4.2 Transformations of the time axis.** Strictly monotonic transformations of the carrier space of an NHPPP yield an NHPPP [19]. Consider an NHPPP with intensity functions $\lambda(t)$ and a strictly monotonic transformation of the time axis $u: t \mapsto \tau$ that is differentiable once almost everywhere. On the transformed time axis the point process is an NHPPP with

**Table 2. Common simulation needs in discrete event simulation.**

| # | Sampling task | Sampled times | Number of sampled events | Example |
|---|---|---|---|---|
| I | Any next event | {} or $\{Z_{(1)}\}$ | 0 or 1 | Single event that may (or may not) occur in the interval: death, progression from Stage I to Stage II cancer. |
| II | Exactly one next event | $\{Z_{(1)}\}$ | 1 | Single event which must occur in the interval: death from any cause in a lifetime-horizon simulation. |
| III | Any and all events | {} or $\{Z_{(1)}, Z_{(2)}, \ldots\}$ | $\geq 0$ | Zero, one, or more events: emergence of one or more bladder tumors. |
| IV | At least one next event | $\{Z_{(1)}, Z_{(2)}, \ldots\}$ | $\geq 1$ | One or more events: emergence of bladder tumors when simulating only patients with bladder tumors. |
| V | Event $i - 1$ given $Z_{(i)}$ | $\{Z_{(i-1)}\}$ | 1 | Find the previous event when simulating conditional on a future event: time of symptom onset given the time of symptom-driven diagnosis; onset of Stage I cancer given progression from Stage I to Stage II cancer. |

All listed tasks involve sampling events over the interval $(a, b]$ with known $\lambda(t)$ or $\Lambda(t)$.

intensity function

$$\rho(\tau) = \lambda(\tau) \left( \frac{du}{dt} \right)^{-1}. \tag{5}$$

This property is important for simulation because

- it motivates the use of another general sampling algorithm (Algorithm 5, "time transformation" or "inversion", [19]): A smart choice for $u$ yields an easy to sample point process. The event times in the original time scale can be obtained as $Z_i = u^{-1}(\zeta_i)$, where $\zeta_i$ is the $i$-th event in the transformed time axis and $u^{-1}$ is the inverse function of $u$.

- given that at least $i$ events have realized in the time interval $(a, b]$, it makes it possible to draw events $Z_{(j)}, j < i$ given event $Z_{(i)}$. This is useful for simulating earlier events conditional on the occurrence of a subsequent event. Choosing $u(t) := Z_{(i)} - t$ makes the time count backwards from $Z_{(i)}$. In this reversed clock we draw as if in forward time exactly $i - 1$ events $\zeta_{(1)}$, $\zeta_{(2)}, \ldots, \zeta_{(i-1)}$. Back transforming yields all preceding events.

Table 2 summarizes the common simulation tasks, such as simulating single events (at most one, exactly one), a series of events (possibly demanding the occurrence of at least one event), or the occurrence of a prior (event $i - 1$ given $Z_{(i)}$). The **nhppp** package implements functions to simulate these tasks for general $\lambda(t)$ or $\Lambda(t)$.

## 3 Sampling the constant rate Poisson process

Sampling the constant rate Poisson process is straightforward. Algorithms 1 and 2 are two ways to sample event times in interval $(a, b]$ with constant intensity $\lambda$. Algorithm 3 describes sampling event times conditional on observing at least $k$ events within the interval of interest.

### 3.1 Sequential sampling

Algorithm 1 samples events sequentially, using the fact that the inter-event times $X_i$ are exponentially distributed with mean $\lambda^{-1}$ [17, par. 4.1]. It involves generation only of exponential random variates, which is cheap on modern hardware. To sample at most $k$ events, change the

condition for the while loop in line 3 to

$$\text{while } t < b \ \& \ |\mathcal{Z}| < k \ \text{do}.$$

The package's `ppp_sequential()` function implements constant-rate sequential sampling that returns a vector with zero or more event times in the interval $[a, b)$. The `range_t` argument is a two-values vector with the bounds $a$, $b$. Setting the optional argument `atmost1` to TRUE from its default value of FALSE returns the first event or an empty vector, depending on whether at least one event is drawn in the interval.

**Algorithm 1** Sequential sampling of events in interval $(a, b]$ with constant intensity $\lambda$.

```
Require: t ∈ (a, b]
 1: t ← a
 2: 𝒵 ← ∅          ▷ 𝒵 is an ordered set
 3: while t < b do   ▷ Up to k earliest points: while t < b & |𝒵| < k do
 4:    X ← X ~ Exponential(λ⁻¹)        ▷ Mean-parameterized
 5:    t ← t + X
 6:    if t < b then
 7:      𝒵 ← 𝒵 ∪ {t}
 8:    end if
 9: end while
10: return 𝒵
```

```
R> library("nhppp")
```

```
R> ppp_sequential(range_t = c(7, 10), rate = 1, atmost1 = FALSE)
```

```
[1] 7.673885 8.650502 9.011229 9.407575
```

**nhppp** functions can accept a user provided random number stream object via the `rng_stream` option.

```
R> library("rstream")
R> S <- new("rstream.mrg32k3a")
R> ppp_sequential(range_t = c(7, 10), rate = 1, rng_stream = S)
[1] 8.793702
```

## 3.2 Sampling using order statistics

**Algorithm 2** Sampling events in interval $(a, b]$ with constant intensity $\lambda$ using order statistics.

```
Require: t ∈ (a, b]
 1: N ← N ~ Poisson(λ(b − a))
 2: t ← a
 3: 𝒵 ← ∅          ▷ 𝒵 is an ordered set
 4: if N > 0 then
 5:    for i ∈ [N] do:
 6:      Uᵢ ← Uᵢ ~ Uniform(0, 1)       ▷ Generate order statistics
 7:      𝒵 ← 𝒵 ∪ {a + (b − a)Uᵢ}
 8:    end for
 9:    𝒵 ← sort(𝒵)
10: end if
11: return 𝒵     ▷ Up to k earliest points: return {Z₍ᵢ₎ | i ≤ k , Z₍ᵢ₎ ∈ 𝒵}
```

Algorithm 2 first draws the number of events in $(a, b]$ from a Poisson distribution. Conditional on the number of events, the event times $Z_i$ are uniformly distributed over $(a, b]$ [17, par. 4.1]. The algorithm returns the order statistics $[Z_{(i)}]$, obtained by sorting the event times $[Z_i]$ in ascending order. It is necessary to generate all event times to generate the order statistics. Thus, to sample at most $k$ event times we should return the earliest $k$ event times, and line 11 of the Algorithm would be changed to

$$\textbf{return } \{Z_{(i)} \mid i \leq k, Z_{(i)} \in \mathcal{Z}\}.$$

The `ppp_orderstat()` function implements constant-rate sampling via the order-statistics algorithm.

```
R> ppp_orderstat(range_t = c(3.14, 6.28), rate = 1/2)
[1] 3.141663 5.700931
```

### 3.3 Sampling conditional on observing at least $m$ events

**Algorithm 3** Sampling with constant intensity $\lambda$ conditional that at least $m$ events occurred in interval $(a, b]$. Relies on generating order statistics analogously to Algorithm 2.

```
Require: t ∈ (a, b]
 1: N ← N ~ TruncatedPoisson_{N≥m}(λ(b − a))  ▷ (m − 1)-truncated Poisson
 2: t ← a
 3: 𝒵 ← ∅           ▷ 𝒵 is an ordered set
 4: if N > 0 then
 5:   for i ∈ [N] do:
 6:     U_i ← U_i ~ Uniform(0, 1)        ▷ Generate order statistics
 7:     𝒵 ← 𝒵 ∪ {a + (b − a)U_i}
 8:   end for
 9:   𝒵 ← sort(𝒵)
10: end if
11: return 𝒵    ▷ Up to k earliest points: return {Z_{(i)} | i ≤ k , Z_{(i)} ∈ 𝒵}
```

Algorithm 3 is used to generate a point process conditional on observing at least $m$ events. For example, if $\lambda$ is the intensity of tumor generation, it can be used to simulate times of tumor emergence among patients with at least one ($m = 1$) tumor. To return the up to $k$ earliest events, we modify line 11 the same way as for Algorithm 2. As an example, in a lifetime simulation we can sample the time of all-cause death by setting in Algorithm 3 $m = 1$, so that at least one event will happen in $(a, b]$, and $k = 1$, to sample only the time of the first event $Z_{(1)}$.

To sample exactly $m$ events, change line 1 of Algorithm 3 to

$$N \leftarrow m.$$

Function `ztppp()` simulates times conditional on drawing at least one event—i.e., setting $m = 1$ in Algorithm 3 to sample from a zero truncated Poisson distribution in line 1.

```
R> ztppp(range_t = c(0, 10), rate = 0.001, atmost1 = FALSE)
[1] 4.411277
```

Function `ppp_n()` simulates times conditional on drawing exactly $m$ events.

```
R> ppp_n(size = 4, range_t = c(0, 10))
[1] 1.762014 2.902897 6.751627 9.733794
```

## 4 The general sampling algorithms used in nhppp

The **nhppp** package uses three well known general sampling algorithms, namely thinning, time transformation or inversion, and order-statistics. These algorithms are efficiently combined to sample from special cases, including cases where the intensity function is a piecewise constant, linear, or log-linear function of time, as described in Section 5.2.

The thinning algorithm works with the intensity function $\lambda(t)$, which is commonly available. The inversion and order statistics algorithms have smaller computational cost than the thinning algorithm, but work with the integrated intensity function $\Lambda(t)$ and its inverse $\Lambda^{-1}(z)$, which may not be available. The generic function `draw()` is a wrapper function that dispatches to specialized functions depending on the provided arguments. It is useful for general tasks but the specialized functions are probably faster.

```
R> l <- function(t) t

R> L <- function(t) 0.5 * t^2

R> Li <- function(z) sqrt(2 * z)

R> draw(

+ lambda = l, lambda_maj = l(10), range_t = c(5, 10),

+ atmost1 = FALSE, atleast1 = FALSE

+ ) |> head(n = 5)
[1] 5.179473 5.374814 5.957391 5.992196 6.101935
R> draw(

+ Lambda = L, Lambda_inv = Li, range_t = c(5, 10),

+ atmost1 = FALSE, atleast1 = FALSE

+ ) |> head(n = 5)
[1] 5.219264 5.230747 5.369646 5.398531 5.618079
```

### 4.1 The thinning algorithm

The thinning algorithm relies on the decomposability of NHPPPs (Section 2.4) and is described in [18]. Let the target NHPPP have intensity function $\lambda(t)$ and $\lambda_*(t) \geq \lambda(t)$ for all $t \in (a, b]$ be a majorizing intensity function. Think of the majorizing function as an easy-to-sample function which is the sum of the intensity of the target point process $\lambda(t)$ and the intensity $\lambda_{reject}(t)$ of its complementary point-process,

$$\lambda_*(t) = \lambda(t) + \lambda_{reject}(t).$$

The acceptance-rejection scheme in Algorithm 4 generates proposal samples with intensity function $\lambda_*(t)$ and stochastically attributes them to the target process (to keep, with probability $\lambda(Z)/\lambda_*(Z)$) or its complement.

**Algorithm 4** The thinning algorithm for sampling from $\lambda(t)$.

```
Require:
    λ_*(t) ≥ λ(t) ∀ t ∈ (a, ]          ▷ majorizing intensity function
    Z_* = {Z_i^* | Z_i^* are samples from λ_*(t)}        ▷ Z_* is an ordered set
 1: N ← |Z_*|
 2: Z ← ∅                                               ▷ Z is an ordered set
 3: if N > 0 then
 4:   for i ∈ [N] do:
 5:     U_i ← U_i ∼ Uniform(0, 1)
 6:     if U_i < λ(Z_(i)^*)/λ_*(Z_(i)^*) then
 7:       Z ← Z ∪ {Z_(i)^*}
 8:     end if
 9:   end for
10: end if
11: return Z      ▷ Up to k earliest points: return {Z_(i) | i ≤ k , Z_(i) ∈ Z}
```

To sample the earliest $k$ points, one can exit the for loop in lines 4–9 when $k$ events have been sampled in line 7, or, alternatively, return the first up to $k$ points in line 11.

A measure of the efficiency of Algorithm 4 is the proportion of samples that are accepted, which is

$$\frac{\int_a^b \lambda(t)\mathrm{d}t}{\int_a^b \lambda_*(t)\mathrm{d}t} \tag{6}$$

on average. Thus, the closer $\lambda_*(t)$ is to $\lambda(t)$, the more efficient the algorithm.

In practice, $\lambda_*(t)$ can be chosen as one of the special cases in Section 5, for which we have fast sampling algorithms. For example, it can be a piecewise constant majorizer. Algorithm A in S1 Appendix can automatically generate a piecewise constant majorizer function for intensity functions that are monotonic and possibly non-continuous or Lipschitz continuous and possibly non-monotonic.

The **nhppp** package has functions that sample from time-varying intensity functions. The first function, draw_intensity(), expects a user-provided linear ($\lambda_*(t) = \alpha + \beta t$) or log-linear ($\lambda_*(t) = e^{\alpha + \beta t}$) majorizer function.

```
R> lambda_fun <- function(t) exp(0.02 * t)
R> draw_intensity(
+ lambda = lambda_fun, # linear majorizer
+ lambda_maj = c(intercept = 1.01, slope = 0.03),
+ exp_maj = FALSE, range_t = c(0, 10)
+ ) |> head(n = 5)

[1] 1.310245 2.094217 2.908682 3.268384 8.007606

R> draw_intensity(
+ lambda = lambda_fun, # log-linear majorizer
+ lambda_maj = c(intercept = 0.01, slope = 0.03),
+ exp_maj = TRUE, range_t = c(0, 10)
+ ) |> head(n = 5)
[1] 0.3406743 0.6079479 0.8441584 2.6424551 3.3185387
```

The second function, `draw_intensity_step()`, expects a user-provided piecewise linear majorizer

$$\lambda_*(t) = \begin{cases} \lambda_1 & \text{for } t \in [a_1, b_1) = [a, b_1), \\ \dots & \\ \lambda_m & \text{for } t \in [a_m, b_m) \text{ with } a_m = b_{m-1}, \\ \dots & \\ \lambda_M & \text{for } t \in [a_M, b_M) = [a_M, b), \end{cases}$$

which is specified as a vector of length $M + 1$ including the points $(a, [b_m]_{m=1}^M)$ and a vector of length $M$ with the values $[\lambda_m]_{m=1}^M$ in each subinterval of $(a, b)$. For example, the following code splits the interval $(0, 10]$ into $M = 10$ subintervals of length one. Because `lambda_fun()` is strictly increasing, its value at the upper bound of each subinterval is the supremum of the interval.

```
R> draw_intensity_step(
+ lambda = lambda_fun,
+ lambda_maj_vector = lambda_fun(1:10), # 1:10 (10 intensity values)
+ times_vector = 0:10 # 0:10 (11 interval bounds)
+ ) |> head(n = 5)
[1] 0.3825378 7.0822941 7.7839779 8.7766992 8.9554954
```

## 4.2 The time transformation or inversion algorithm

Algorithm 5 implements the time transformation or inversion algorithm from [19] and [17, par. 4.2]. As mentioned in Section 2.4, strictly monotonic transformations of the carrier space (here, time) of a Poisson point process yield another Poisson Point Process. In Eq (5), choosing the transformation $\tau = u(t) = \Lambda(t)$, so that $\frac{du(t)}{dt} = \lambda(t)$, results in $\rho(\tau) = 1$.

This means (proof sketched in [17, par. 4.2]) that we can sample points from a Poisson point process with intensity one over the interval $(\tau_a, \tau_b] = (\Lambda(a), \Lambda(b)]$. Via a similar argument, we transform event times sampled on the transformed scale back to the original scale using $g(t) = \Lambda^{-1}(\tau)$. The transformations $u(\cdot), g(\cdot)$ are not unique—at least up to the group of affine transformations.

Function `draw_cumulative_intensity_inversion()` works with a cumulative intensity function $\Lambda(t)$ and its inverse $\Lambda^{-1}(z)$, if available. If the inverse function is not available (argument `Lambda_inv = NULL`), the Brent bisection algorithm is used to invert $\Lambda(t)$ numerically, at a performance cost [20].

```
R> Lambda_fun <- function(t) 50 * exp(0.02 * t) - 50
R> Lambda_inv_fun <- function(z) 50 * log((z + 50) / 50)
R> draw_cumulative_intensity_inversion(
+ Lambda = Lambda_fun,
+ Lambda_inv = Lambda_inv_fun,
+ range_t = c(5, 10.5), + range_L = Lambda_fun(c(5, 10.5))
+ ) |> head(n = 5)
[1] 6.458937 7.608496 9.060817 9.566278 10.076889
```

**Algorithm 5** The time transformation or inversion algorithm for sampling given $\Lambda(t)$, $\Lambda^{-1}(z)$ [17, 19]. The notation PoissonProcess$_1$ indicates sampling event times from a constant rate one Poisson point process.

```
Require: Λ(t), Λ⁻¹(z), t ∈ (a, b]              ▷ Λ⁻¹(z) possibly numerically
  1: τ_a ← Λ(a),  τ_b ← Λ(b)
  2: C ← C ∼ PoissonProcess₁(τ_a,τ_b)         ▷ From Algorithm 1 (or 3 for
                                                 conditional sampling)
  3: Z ← Λ⁻¹(C) ▷ Λ⁻¹(·) as set function, meant elementwise
  4: return Z
```

## 4.3 The order statistics algorithm

The general order statistics algorithm (Algorithm 6) is a direct generalization of Algorithm 2. It first draws the number $N$ of realized events. Conditional on $N$

$$U_{(i)} = \frac{\Lambda(Z_{(i)}) - \Lambda(a)}{\Lambda(b) - \Lambda(a)} \sim \text{Uniform}(0, 1),$$

$$Z_{(i)} = \Lambda^{-1}(\Lambda(a) + U_{(i)}(\Lambda(b) - \Lambda(a))), \tag{7}$$

as discussed in [18]. Algorithm 6 makes the above explicit.

**Algorithm 6** The order statistics algorithm for sampling from an NHPPP given $\Lambda(t)$, $\Lambda^{-1}(z)$.

```
Require: Λ(t), Λ⁻¹(z), t ∈ (a, b]              ▷ Λ⁻¹(z) possibly numerically
  1: N ← N ∼ Poisson(Λ(b) − Λ(a))
  2: t ← a
  3: Z ← ∅           ▷ Z is an ordered set
  4: if N > 0 then
  5:   for i ∈ [N] do:
  6:     U_i ← U_i ∼ Uniform(0, 1)              ▷ Generate order statistics
  7:     Z ← Z ∪ {Λ⁻¹(Λ(a) + U_i(Λ(b) − Λ(a)))}
  8:   end for
  9:   Z ← sort(Z)
 10: end if
 11: return Z           ▷ Up to k earliest points: return
         {Z_(i) | i ≤ k , Z_(i) ∈ Z}
```

Sampling up to $k$ earliest points means returning the up to $k$ earliest event times. If $\Lambda(t)$ is a positive linear function of time, $\lambda$ is constant and Algorithm 6 becomes Algorithm 2.

To sample conditional on observing at least $m$ events in the interval $(a, b]$ see Algorithm B in S2 Appendix.

$$N \leftarrow N \sim \text{TruncatedPoisson}_{N \geq m}(\Lambda(b) - \Lambda(a)).$$

Function `draw_cumulative_intensity_orderstats()` works with a cumulative intensity function $\Lambda(t)$ and its inverse $\Lambda^{-1}(z)$, if available. Function `ztdraw_cumulative_intensity()` conditions that at least one event is sampled in the interval. As above, if the inverse function is not available (argument `Lambda_inv = NULL`), the Brent bisection algorithm is used to invert $\Lambda(t)$ numerically, at a performance cost.

```
R> draw_cumulative_intensity_orderstats(
+ Lambda = Lambda_fun,
+ Lambda_inv = Lambda_inv_fun,
+ range_t = c(4.1, 7.6)
+ )
```

```
[1] 5.091581 5.526070 5.601576 5.762498 6.495684
R> ztdraw_cumulative_intensity(
+ Lambda = Lambda_fun,
+ Lambda_inv = Lambda_inv_fun,
+ range_t = c(4.1, 7.6)
+ )
[1] 5.063676 6.682454 6.749162 6.926164 7.298342
```

## 5 Special cases

The **nhppp** package implements several special cases where the intensity function $\lambda(\cdot)$, the integrated intensity function $\Lambda(\cdot)$, and its inverse $\Lambda^{-1}(\cdot)$ have straightforward analytical expressions.

### 5.1 Sampling a piecewise constant NHPPP

Functions `draw_sc_step()` and `draw_sc_step_regular()` sample piecewise constant intensity functions based on Algorithm 5. The first can work with unequal-length subintervals $(a_m, b_m]$. The second results in a small computational time improvement when all subintervals are of equal length.

```
R> draw_sc_step(
+ lambda_vector = 1:5, times_vector = c(0.5, 1, 2.4, 3.1, 4.9, 5.9),
+ atmost1 = FALSE, atleast1 = FALSE
+ ) |> head(n = 5)
[1] 0.84251117 1.3281115 2.3309443 2.6794560 2.7939130
R> draw_sc_step_regular(
+ lambda_vector = 1:5, range_t = c(0.5, 5.9), atmost1 = FALSE,
+ atleast1 = FALSE
+ ) |> head(n = 5)
[1] 2.058468 2.100620 2.508954 3.125179 3.604882
```

Function `vdraw_sc_step_regular()` is a vectorized version of `draw_sc_step_regular()`. It returns a matrix with one event series per row, and as many columns as the maximum number of events across all draws.

```
R> vdraw_sc_step_regular(
+ lambda_matrix = matrix(runif(20), ncol = 5), range_t = c(1, 4),
+ atmost1 = FALSE
+ )      [,1]      [,2]      [,3]      [,4]
[1,] 2.304123  2.802767       NA        NA
[2,] 2.990953       NA        NA        NA
[3,] 1.840374  2.134357  3.784424  3.816034
[4,] 2.136138  2.703826  3.269631        NA
```

The corresponding functions that return at least one event in the interval are `ztdraw_sc_step()`, `ztdraw_sc_step_regular()`, and `vztdraw_sc_step_regular()`.

## 5.2 Sampling NHPPPs with linear and log-linear intensities

Functions `draw_sc_linear()` and `ztdraw_sc_linear()` sample zero or more and at least one event, respectively, from NHPPPs with linear intensity functions. An optional argument (`atmost1`) returns the first event only.

$$\lambda(t) = \begin{cases} \alpha + \beta t & \text{for } t \in [a, b), t > -\dfrac{\alpha}{\beta} \\ 0 & \text{otherwise} \end{cases}.$$

```
R> draw_sc_linear(alpha = 3, beta = -0.5, range_t = c(0, 10)) |> head
(n = 5)
[1] 0.3327657 0.4270154 0.5804320 0.6935027 0.9832093
R> ztdraw_sc_linear(alpha = 0.5, beta = 0.2, range_t = c(9.999, 10))
[1] 9.999757
```

An analogous set of functions (`[nhppp|ztnhppp]_sc_loglinear()`) samples from log-linear intensity functions

$$\lambda(t) = \begin{cases} e^{\alpha + \beta t} & \text{for } t \in [a, b) \\ 0 & \text{otherwise} \end{cases}.$$

The sampling algorithm is a variation of Algorithm 5, as described in [21]. Example usage follows.

```
R> draw_sc_loglinear(alpha = 1, beta = -0.02, range_t = c(8, 10))
[1] 8.028806 8.128887 8.457669 8.483558 8.498647 8.503109 8.522725
[8] 8.665979 8.671737 8.978065 8.981105 9.493691 9.815000 9.909167
R> ztdraw_sc_loglinear(alpha = 1, beta = -0.02, range_t = c(9, 10))
[1] 9.038160 9.075722 9.238302
```

## 6 Comparisons with other R packages

Table 3 lists five R packages that simulate from NHPPPs, including **nhppp**. We did not consider research code that is not an R package in the Comprehensive R Archive Network or is developed in other languages. For example, we do not run comparisons with the R and Python code for sampling from piecewise constant NHPPPs with regular time intervals in Garibay *et al* [22]. (Their code corresponds to the `vdraw_sc_step_regular()` function in **nhppp**.)

Package **reda** [23] focuses on recurrent event data analysis and can simulate NHPPPs with the inversion and thinning algorithms using the `simEvent()` function. It can take function object arguments for $\lambda(t)$. When using the thinning algorithm, it takes a constant majorizer. For the inversion algorithm, it approximates $\Lambda(t)$ and its inverse numerically, at a computational cost.

Package **simEd** [24] includes various functions for simulation education. Function `thinning()` implements the homonymous algorithm for drawing points from an NHPPP. Users can specify the intensity function and a piecewise constant or linear majorizer function.

**Table 3. NHPPP generation in R packages.**

| R package | Function | Algorithms (inputs) | | | Sample only earliest event | Custom RNG | Simulate given $N > 0$ | Vectorized functions |
|---|---|---|---|---|---|---|---|---|
| | | Thinning | Inversion | statistics | | | | |
| **nhppp** | [see text] | $\lambda(t)$, $\lambda_*(t)$ | $\Lambda(t)$, $\Lambda^{-1}(z)$ | $\Lambda(t)$, $\Lambda^{-1}(z)$ | Yes | **rstream** objects | Yes | For piecewise constant intensity |
| **reda** | `simEvent()` | $\lambda(t)$, $\lambda_*$ constant | $\lambda(t)$ (no $\Lambda(t)$, $\Lambda^{-1}(z)$) | No | Yes | No | No | No |
| **simEd** | `thinning()` | $\lambda(t)$, $[\lambda_{*m}]_{m=1}^{M}$ | No | No | No | No | No | No |
| **IndTestPP** | `simNHPc()` | $[\lambda_m]_{m=1}^{M}$, $\lambda_*$ constant | $[\lambda_m]_{m=1}^{M}$ (no $\Lambda(t)$, $\Lambda^{-1}(z)$) | No | No | No | No | No |
| **NHPoisson** | `simNHP.fun ()` | No | $\lambda(t)$, (no $\Lambda(t)$, $\Lambda^{-1}(z)$) | No | No | No | No | No |

RNG: random number generator object.

Package **IndTestPP** [25] provides a framework for exploring the dependence between two or more realizations of point processes. It includes the ancillary function `simNHPc()` for simulating NHPPPs with the inversion or thinning algorithms. The function's argument is a piecewise constant approximation of the intensity function via a vector of evaluations, each corresponding to unit length subintervals. This resolution may not be adequate to simulate processes that change fast over a unit time interval.

Package **NHPoisson** [26, 27] fits NHPPP models to data and is not really geared towards mathematical simulation. Its `simNHP.fun()` function provides the ability for simulation-based inference via an implementation of the inversion algorithm. This function is designed to work with the package's inference machinery and is not practical to use for simulation, because the user has no direct control over the function's rescaling of the time axis.

The claimed advantage of **nhppp** over the existing packages is that

- it samples from the target NHPPP and not from a numerical approximation thereof, e.g., as **IndTestPP** does.

- It can sample conditional on observing at least one event in the interval, which no other package implement.

- It accepts user-provided random number stream objects, which is useful for implementing simulation variance reduction techniques such as common random numbers [28] and antithetic variates [29].

- It is fast and memory efficient, both for the non-vectorized functions that are implemented in native R and for the vectorized functions that use C++ plugins via the **Rcpp** package [30]. **nhppp** has specialized functions to leverage additional information about the point process, such as $\Lambda(t)$, $\Lambda^{-1}(z)$, when available, which can result in faster simulation use the cumulative intensity function and its inverse, often at a computational speed advantage.

## 7 Illustrations

Depending on the application, we may have access to the intensity function or the integrated intensity function. We compared the R packages in Table 3 for sampling from a non-

**Table 4. Simulation metrics for the number of counts.**

| Metric | Definition | Description |
|---|---|---|
| Bias in mean | $B_\mu = \frac{1}{J}\sum_j n_j - N$ | Mean difference from target in the number of counts. |
| Relative bias in mean | $B_{\mu,rel} = \frac{B_\mu}{N}$ | Mean proportional difference from target in the number of counts. |
| Bias in variance | $B_V = \frac{1}{J}\sum_j \left(n_j - \frac{1}{J}\sum_j n_j\right)^2 - V$ | Mean difference from target in variance of counts. |
| Relative bias in variance | $B_{V,rel} = \frac{B_V}{V}$ | Mean proportional difference from target in variance of counts. |
| Equal-tailed $p$% confidence interval bounds | $n_{[p/2]}, n_{[1-p/2]}$ | Quantiles of the empirical distribution of counts. |
| Goodness of fit $p$ value | Statistic $\sum_x \frac{(O_x - E_x)^2}{E_x} \sim \chi^2_{U-L+1}$ | Left-tail $p$ value. $p$ values near 1 imply good fit. |
| Wasserstein-1 distance | $W_1$, the smallest rearrangement of probability mass so that one distribution matches the other. | $W_1 = 0$ implies good fit |
| $p$ value for $W_1 \neq 0$ | Asymptotic theory $p$ value | Two-sided $p$ value. $p$ values near 1 imply good fit. |

In the Table, $j \in [J]$ indexes simulations, $n_j$ is the number of counts in simulation $j$, $N = \Lambda(6\pi) - \Lambda(0)$ is the theoretical mean number of counts, and $V = \Lambda(6\pi) - \Lambda(0) = N$ the theoretical variance. The lower and upper bounds of an equal-tailed $p$% confidence interval, $p \in \{95, 90, 75, 50\}$, are denoted with $n_{[p/2]}, n_{[1-p/2]}$, respectively. For the goodness of fit, we created bins $[0, L), [L, L+1), \ldots, [U, \infty)$, where $L, U$ are the 0.001 and 0.999 percentiles of the Poisson distribution with parameter $\Lambda(6\pi) - \Lambda(0)$. We indexed bins with $x \in \{1, \ldots, U - L + 2\}$. The goodness of fit statistic contrasts the observed ($O_x$) versus expected ($E_x$) numbers of events over the bins and it is compared with a $\chi^2_{U-L+1}$ distribution to obtain a $p$ value.

monotonic and highly non-linear intensity function for which the integrated intensity function can be derived analytically.

## 7.1 The target NHPPP to be simulated

Consider the example

$$
\begin{aligned}
\lambda(t) &= e^{rt}(1 + \sin\ wt), \\
\Lambda(t) &= \frac{e^{rt}(r\ \sin\ wt - w\ \cos\ wt) + w}{r^2 + w^2} + \frac{e^{rt} - 1}{r}
\end{aligned}
\tag{8}
$$

of a sinusoidal intensity function $\lambda(t)$ scaled to have an exponential amplitude and one of its antiderivatives $\Lambda(t)$, with such a constant term that $\Lambda(0) = 0$. For the numerical study we set $r = 0.2$, $w = 1$, and $t \in (0, 6\pi]$. There is no analytic inverse function for this example. However, we can precompute `Li()`, a good numerical approximation to $\Lambda^{-1}(z)$. We will use it in Section 7.5 to compare the time performance of functions that use the inversion and order statistics algorithms when $\Lambda^{-1}$ is available versus not.

```
R> l <- function(t) (1 + sin(t)) * exp(0.2 * t)
R> L <- function(t) {
+ exp(0.2 * t) * (0.2 * sin(t) - cos(t)) / 1.04 +
+ exp(0.2 * t) / 0.2 - 4.038462
+ }
```

```
R> Li <- approxfun(
+ x = L(seq(0, 6 * pi, 10^-3)),
+ y = seq(0, 6 * pi, 10^-3), rule = 2
+ )
```

Fig 1 graphs the intensity function and three majorizing functions over the interval of interest, which will be needed for the thinning algorithm.

The first, $\lambda_{*a}(t) = 43.38$, shown as a dashed blue line, is is a constant majorizer equal to the maximum of the intensity function. A constant majorizer may be a practical choice when only an upper bound is known for $\lambda(t)$. From (6), the analytic efficiency of the thinning algorithm using this majorizer is 0.209.

The second, $\lambda_{*b}(t)$, shown as a thin black line, is a piecewise constant envelope generated automatically from Algorithm A in S1 Appendix with 20 equal-length subintervals and Lipschitz cone coefficient $K = 52.05$. We set $K$ equal to the maximum value of $|\frac{d\lambda(t)}{dt}|$ in the interval, attained at $6\pi$. The analytic efficiency of the thinning algorithm using this majorizer is 0.245.

The third, $\lambda_{*c}(t)$, shown as a thicker black line, is a tighter piecewise constant majorizer with the same 20 equal-length subintervals that is constructed by finding a least upper bound in each subinterval. The analytic efficiency of the thinning algorithm with the third majorizer is 0.718.

## 7.2 Simulation functions and algorithms

We sampled series of events from the target NHPPP using the packages and functions listed in Table 3. We repeated the sampling $10^4$ times, recording all simulated points (event times). We

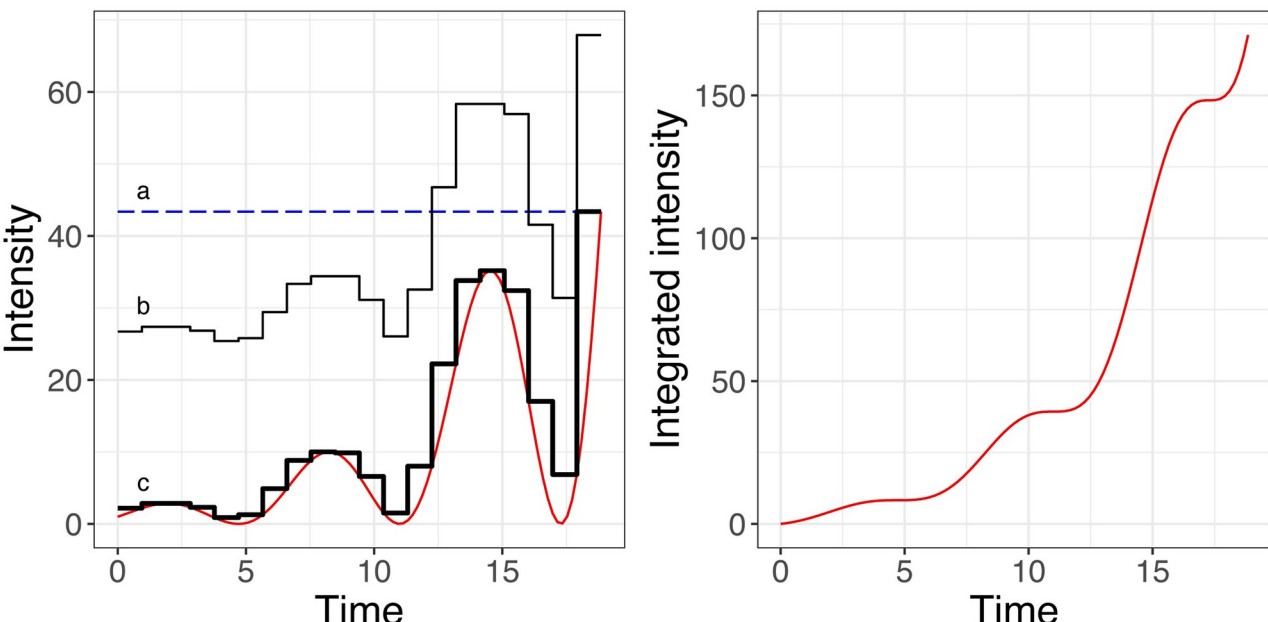

**Fig 1.** The $\lambda(t)$ (left) and $\Lambda(t)$ used in the illustration. Also shown three majorizing functions (left panel, marked a, b, c) that are used with the thinning algorithm in the analyses.

also recorded the median computation time for drawing one series of events with single-threaded computation on modern hardware.

From the **nhppp** package we use

1. two functions that take as argument the intensity function and are based on Algorithm 4 (thinning): `draw_intensity()`, which uses linear majorizers such as $\lambda_{*a}$, and `draw_intensity_step()`, which uses piecewise constant majorizers such as $\lambda_{*b}$ and $\lambda_{*c}$ in the example.

2. Function `draw_cumulative_intensity_inversion()`, which takes as argument the cumulative intensity function $\Lambda(t)$ and is based on Algorithm 5 (time transformation/inversion), and

3. function `draw_cumulative_intensity_orderstats()`, which also uses $\Lambda(t)$ and is based on Algorithm 6 (order statistics).

Regarding the other R packages in Table 3, we used all except for **NHPoisson**, whose simulation function is tailored to supporting simulation based inference for data analysis and is not practical to use as a standalone function. (Its implementation does not allow the user to control the scaling of the time axis in a practical way.) However, its source code/algorithm is very similar to that of the **IndTestPP** simulation function, which is developed by the same authors.

We used the metrics in Table 4 to assess simulation performance with each function. We compared the empirical versus the simulated distributions of number of events and event times over $J = 100$ simulation runs.

## 7.3 Simulation performance with respect to number of events

We calculated the absolute and relative bias in the first two moments of the empirical distribution in the counts of events, the bounds of equal-tailed confidence intervals at the 95, 90, 75, and 50 percent levels, a $\chi^2$-distributed goodness of fit statistic and its *p*-value, and the Wasserstein-1 distance $W_1$ between the empirical and the theoretical count distributions and the asymptotic one sided *p* value to reject whether $W_1 = 0$ according to [31]. $W_1$ is the smallest mass that has to be redistributed so that one distribution matches the other. $W_1$ is equal to the unsigned area between the cumulative distribution functions of the compared distributions. For example, $W_1 = 5.25$ means that the mass that must be moved to transform one density to the other is no less than 5.25 counts and a $W_1 = 0$ implies perfect fit.

The results for the **nhppp** functions in Fig 2 and Table 5 suggest excellent simulation performance.

The respective results for the R packages are in Fig 3 and Table 6. The simulation performance with the **reda** functions is excellent. Performance with **simEd** and **IndTestPP** functions depends on the adequacy with which they approximate the target density. In this example, the approximation accuracy is not ideal for either package, but is somewhat worse for **IndTestPP**.

## 7.4 Event times

We compared the theoretical and empirical distribution of event times for all $J = 10^4$ event time draws. We calculated a goodness of fit statistic by binning realized times in 70 bins and its *p* value, by comparing the statistic against the $\chi^2_{69}$ distribution. We also calculated the $W_1$ distance between these distributions and its associated *p* value.

Fig 4 and Table 7 indicate excellent simulation performance with the **nhppp** functions.

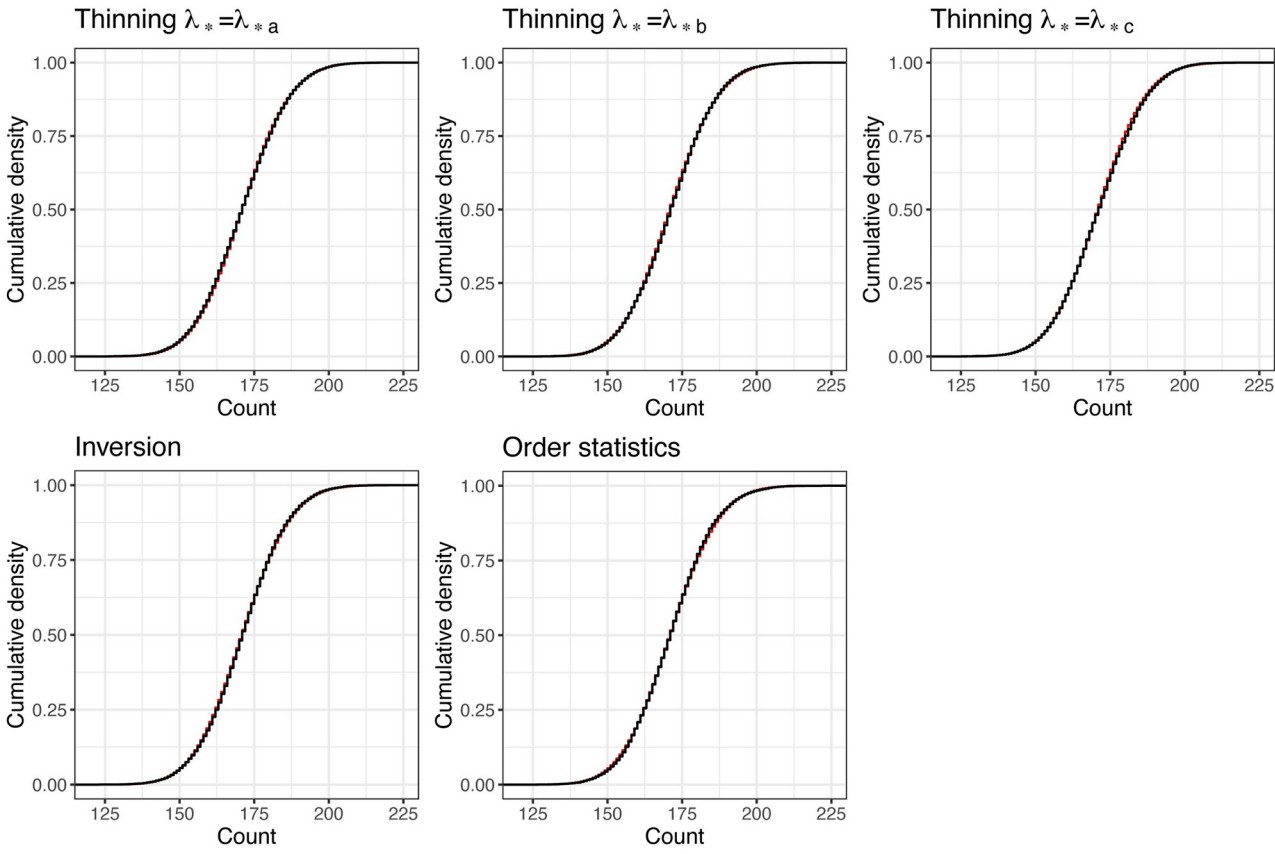

**Fig 2. Theoretical (red) and empirical (black) cumulative distribution functions for event counts in the illustration example with nhppp functions.** The unsigned area between the theoretical and empirical curves equals the Wasserstein-1 distance in Table 5.

**Table 5. Simulated total number of events with nhppp functions for the illustration example.**

| | Thinning $\lambda_*=\lambda_{*a}$ | Thinning $\lambda_*=\lambda_{*b}$ | Thinning $\lambda_*=\lambda_{*c}$ | Inversion | Order statistics |
|---:|---|---|---|---|---|
| Sample mean | 171.057 | 171.257 | 171.322 | 171.193 | 171.131 |
| $B_\mu$ | -0.078 | 0.122 | 0.187 | 0.058 | -0.004 |
| $B_{\mu,rel}$ | -0.045 | 0.071 | 0.109 | 0.034 | -0.002 |
| Sample variance | 175.015 | 168.218 | 173.918 | 166.950 | 166.933 |
| $B_V$ | 3.880 | -2.917 | 2.783 | -4.185 | -4.201 |
| $B_{V,rel}$ | 2.267 | -1.704 | 1.626 | -2.445 | -2.455 |
| Goodness of fit, $\chi^2$ [$p$ value] | 0.145 [1.000] | 0.160 [1.000] | 0.117 [1.000] | 0.384 [1.000] | 0.229 [1.000] |
| $W_1$ [$p$ value] | 0.194 [1.000] | 0.189 [1.000] | 0.231 [0.997] | 0.195 [1.000] | 0.187 [1.000] |
| Equal tail 95% CI = [146, 197] | [146, 197] | [146, 197] | [146, 197] | [146, 197] | [146, 197] |
| Equal tail 90% CI = [150, 193] | [150, 193] | [150, 193] | [150, 193] | [150, 193] | [150, 193] |
| Equal tail 75% CI = [156, 186] | [156, 186] | [156, 186] | [156, 187] | [156, 186] | [156, 186] |
| Equal tail 50% CI = [162, 180] | [162, 180] | [162, 180] | [162, 180] | [162, 180] | [162, 180] |

Equal tail $p$% CI: a confidence interval whose bounds are the $p/2$ and $(1 − p/2)$ count percentiles of the respective cumulative distribution function.

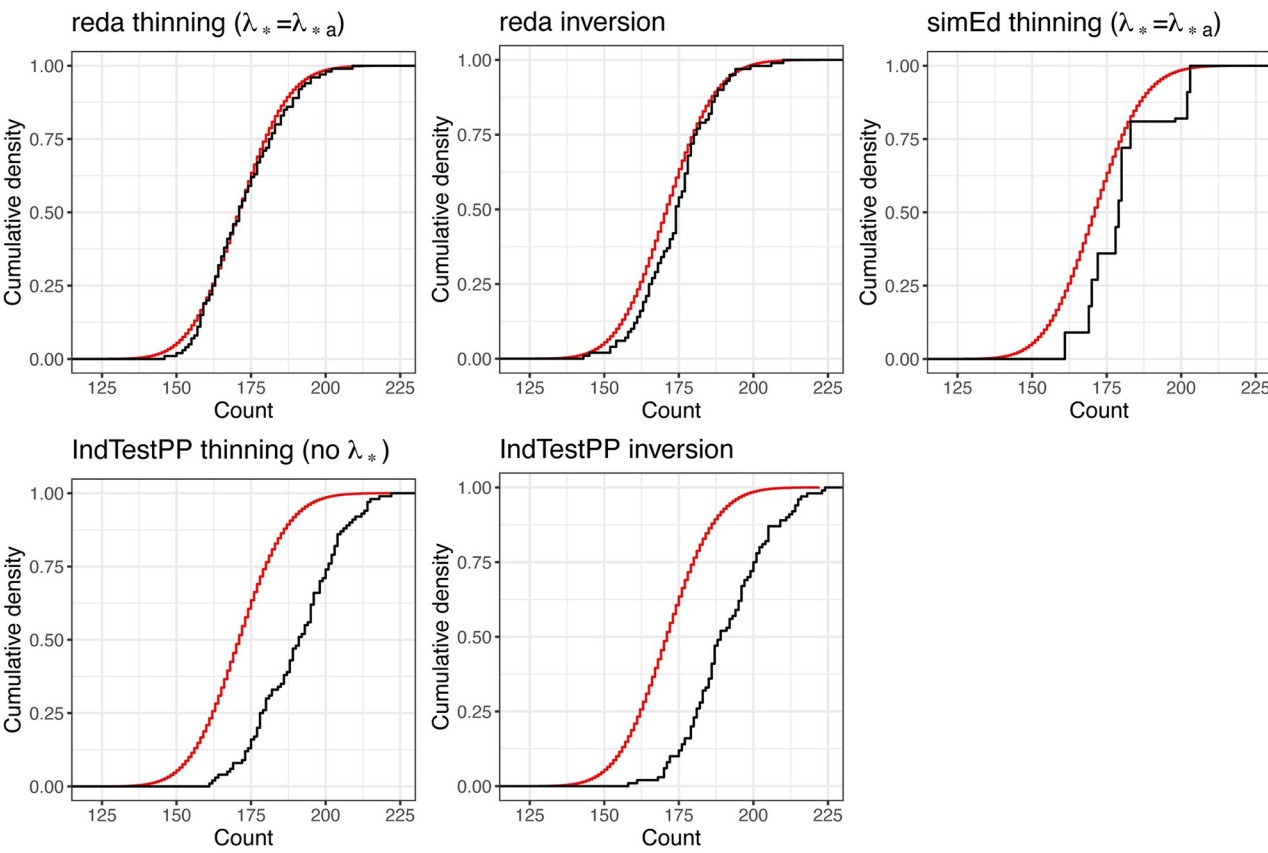

**Fig 3. Theoretical (red) and empirical (black) cumulative distribution functions for event counts in the illustration example with the R packages in Table 3.** The unsigned area between the theoretical and empirical curves equals the Wasserstein-1 distance in Table 5.

Fig 5 and Table 8 indicate excellent simulation performance with the **reda** functions. The simulation performance with the **simEd** and **IndTestPP** functions, which rely on approximations, is not as good.

### 7.5 Time performance

**7.5.1 Time performance of non-vectorized functions.** To indicate time performance, we benchmarked functions by recording execution times when drawing a series of points (Fig 6). We also benchmarked functions for drawing the first-occurring event, because **nhppp** functions can sample the first time more efficiently when the inversion algorithm is used (Fig 7).

We provided functions with the arguments they need to run fastest. For example, functions that use the inversion or order statistics algorithm execute faster when the inverse function $\Lambda^{-1}(z)$ is provided, rather than numerically calculated, as shown in both Figures for the **nhppp** package. (Functions in other packages do not take $\Lambda(t)$ and $\Lambda^{-1}(z)$ arguments.) The fastest functions are **nhppp** functions that rely on the inversion or order statistics algorithms given $\Lambda^{-1}(z)$.

According to (6), the thinning algorithm has higher efficiency, and is expected to execute faster, for majorizer functions that envelop the intensity function more closely. Observe that $\lambda_{*a} \succ \lambda_{*c}$ and $\lambda_{*b} \succ \lambda_{*c}$ in Fig 1. As expected, the execution times are indeed shorter for majorizer 'c' compared to 'b' in Figs 6 and 7. However, the execution times are longer with majorizer

**Table 6. Simulated total number of events with the R packages of Table 3 for the illustration example.**

| | reda thinning, $\lambda_* = \lambda_{*a}$ | reda inversion | simEd thinning, $\lambda_* = \lambda_{*a}$ | IndTestPP thinning, no $\lambda_*$ | IndTestPP inversion |
|---|---|---|---|---|---|
| Sample mean | 172.430 | 174.170 | 179.910 | 190.490 | 191.030 |
| $B_\mu$ | 1.295 | 3.035 | 8.775 | 19.355 | 19.895 |
| $B_{\mu,rel}$ | 0.757 | 1.774 | 5.128 | 11.310 | 11.626 |
| Sample variance | 168.429 | 145.193 | 155.355 | 194.838 | 191.484 |
| $B_V$ | -2.705 | -25.942 | -15.779 | 23.704 | 20.349 |
| $B_{V,rel}$ | -1.581 | -15.159 | -9.220 | 13.851 | 11.891 |
| Goodness of fit, $\chi^2$ [$p$ value] | 6.830 [1.000] | 10.720 [1.000] | 67.482 [0.994] | 226.107 [<0.001] | 237.199 [<0.001] |
| $W_1$ [$p$ value] | 1.453 [0.256] | 3.083 [0.112] | 8.856 [<0.001] | 19.356 [0.086] | 19.896 [0.170] |
| Equal tail 95% CI = [146, 197] | [152, 199] | [152, 196] | [161, 203] | [163, 214] | [168, 217] |
| Equal tail 90% CI = [150, 193] | [154, 195] | [154, 192] | [161, 203] | [167, 214] | [170, 215] |
| Equal tail 75% CI = [156, 186] | [158, 189] | [161, 187] | [169, 202] | [174, 205] | [176, 207] |
| Equal tail 50% CI = [162, 180] | [162, 181] | [165, 180] | [170, 183] | [178, 201] | [181, 200] |

Equal tail $p$% CI: a confidence interval whose bounds are the $p/2$ and $(1 - p/2)$ count percentiles of the respective cumulative distribution function.

'c' compared to 'a' because `draw_intensity()`, the function that uses constant majorizers, and `draw_intensity_step()`, the function that use piecewise constant majorizers, are implemented differently. `draw_intensity()` happens to be faster in this example, but this is not always true.

In **nhppp**, functions that use the inversion or order statistics algorithms can exit earlier when only the first event is requested. This is not possible, however, for the thinning algorithm. This efficiency does not appear to be implemented in the other packages.

**7.5.2 Time performance of vectorized functions.** In R, 'vectorized' computation, where operations are done in columns, is faster than using `for` loops or `apply()` functions. As shown in Table 1, **nhppp** includes vectorized functions for sampling from (i) piecewise constant intensity functions, using `[vdraw|vztdraw]_sc_step_regular()`; and (ii) general intensity functions, using `[vdraw|vztdraw]_intensity_step_regular()`.

We compared the execution speed of non-vectorized and vectorized functions for sampling $10^5$ times from the piecewise constant 'b' majorizer ($\lambda_{*b}$) in Fig 1. The expected number of events with $\lambda_{*b}$ in $(0, 6\pi]$ is 741.97. When drawing only the earliest event, the vectorized function is approximately 113 times faster than the non-vectorized function (median $59ms$ versus $6717ms$ over $10^5$ simulations). When drawing all events, the vectorized function is approximately 1.4 times faster than the non-vectorized function (median $36.55s$ versus $50.97s$ over $10^5$ simulations). The reason that the difference in speed attenuates is that the current implementation of the vectorized functions does not use sparse matrices to store samples, which introduces inefficiencies the expected number of samples becomes larger.

## 8 Summary and next developments

The **nhppp** facilitates the simulation of NHPPPs from time-varying intensity or cumulative intensity functions. Its claim is that it (i) simulates correctly from a target density, not just from an approximation; (ii) samples conditional on observing at least one event in an interval; (iii) accomodates user provided random number stream objects; and (iv) is fast. The current version includes one vectorized function for sampling from regular-spaced piecewise constant intensity functions. In future releases we will further optimize execution speed and memory usage.

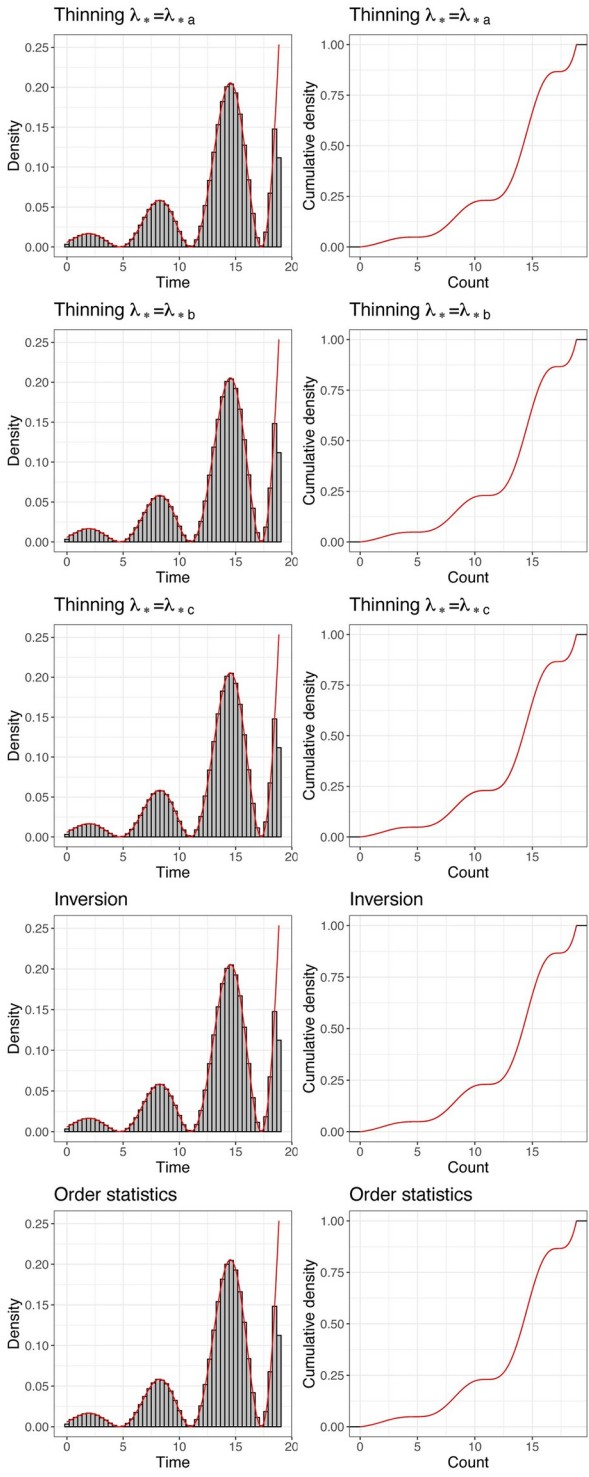

**Fig 4. Simulated event times with nhppp.** Left column: histogram (gray) and theoretical distribution (red) of event times; right column: empirical (black) and theoretical (red) cumulative distribution function. The unsigned area between the empirical and cumulative distribution functions is the $W_1$ distance in Table 7.

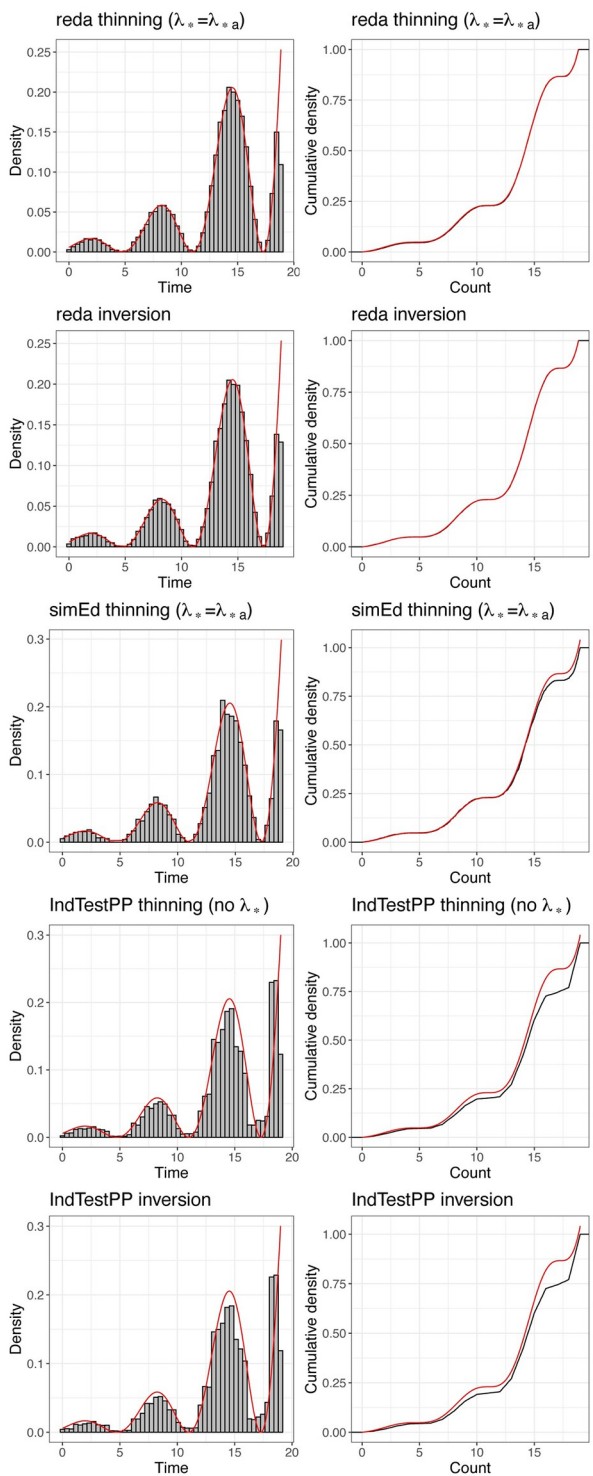

**Fig 5. Simulated event times with the R packages in Table 3.** Left column: histogram (gray) and theoretical distribution (red) of event times; right column: empirical (black) and theoretical (red) cumulative distribution function. The unsigned area between the empirical and cumulative distribution functions is the $W_1$ distance in Table 8.

**Table 7. Goodness of fit of simulated event times with nhppp functions for the example.**

| | Goodness of fit, $\chi^2$ [$p$ value] | $W_1$ [$p$ value] |
|---|---|---|
| Thinning $\lambda_* = \lambda_{*a}$ | 0.004 [1.000] | 0.396 [1.000] |
| Thinning $\lambda_* = \lambda_{*b}$ | 0.004 [1.000] | 0.361 [1.000] |
| Thinning $\lambda_* = \lambda_{*c}$ | 0.004 [1.000] | 0.338 [1.000] |
| Inversion | 0.004 [1.000] | 0.347 [1.000] |
| Order statistics | 0.004 [1.000] | 0.350 [1.000] |

**Table 8. Goodness of fit of simulated event times with R functions in Table 3.**

| | Goodness of fit, $\chi^2$ [$p$ value] | $W_1$ [$p$ value] |
|---|---|---|
| **reda** thinning ($\lambda_* = \lambda_{*a}$) | 0.012 [1.000] | 0.356 [1.000] |
| **reda** inversion | 0.010 [1.000] | 0.354 [1.000] |
| **simEd** thinning ($\lambda_* = \lambda_{*a}$) | 0.028 [1.000] | 0.338 [0.990] |
| **IndTestPP** thinning (no $\lambda_*$) | 0.460 [1.000] | 2.152 [0.930] |
| **IndTestPP** inversion | 0.490 [1.000] | 2.372 [0.927] |

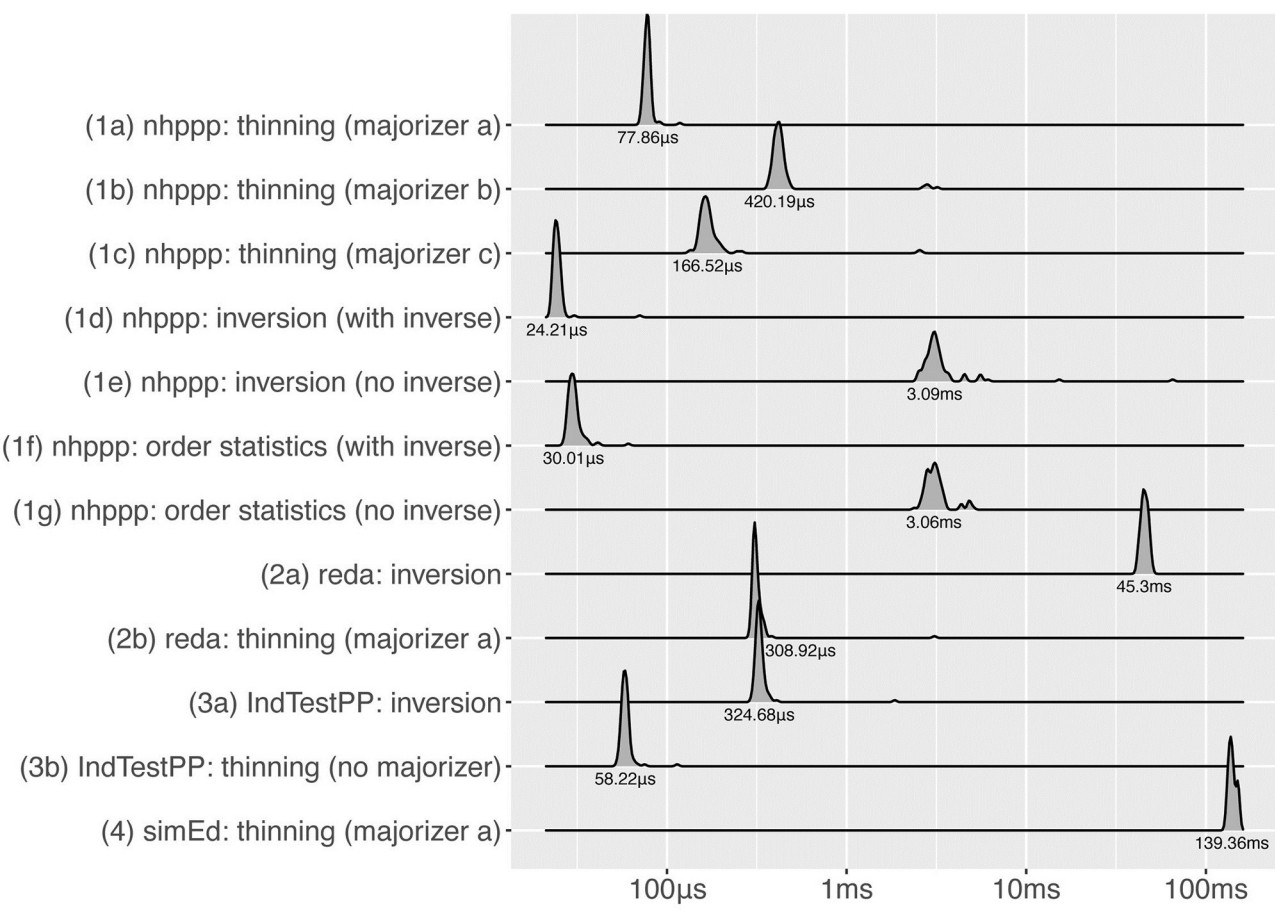

**Fig 6. Computation times when drawing all events in interval.**

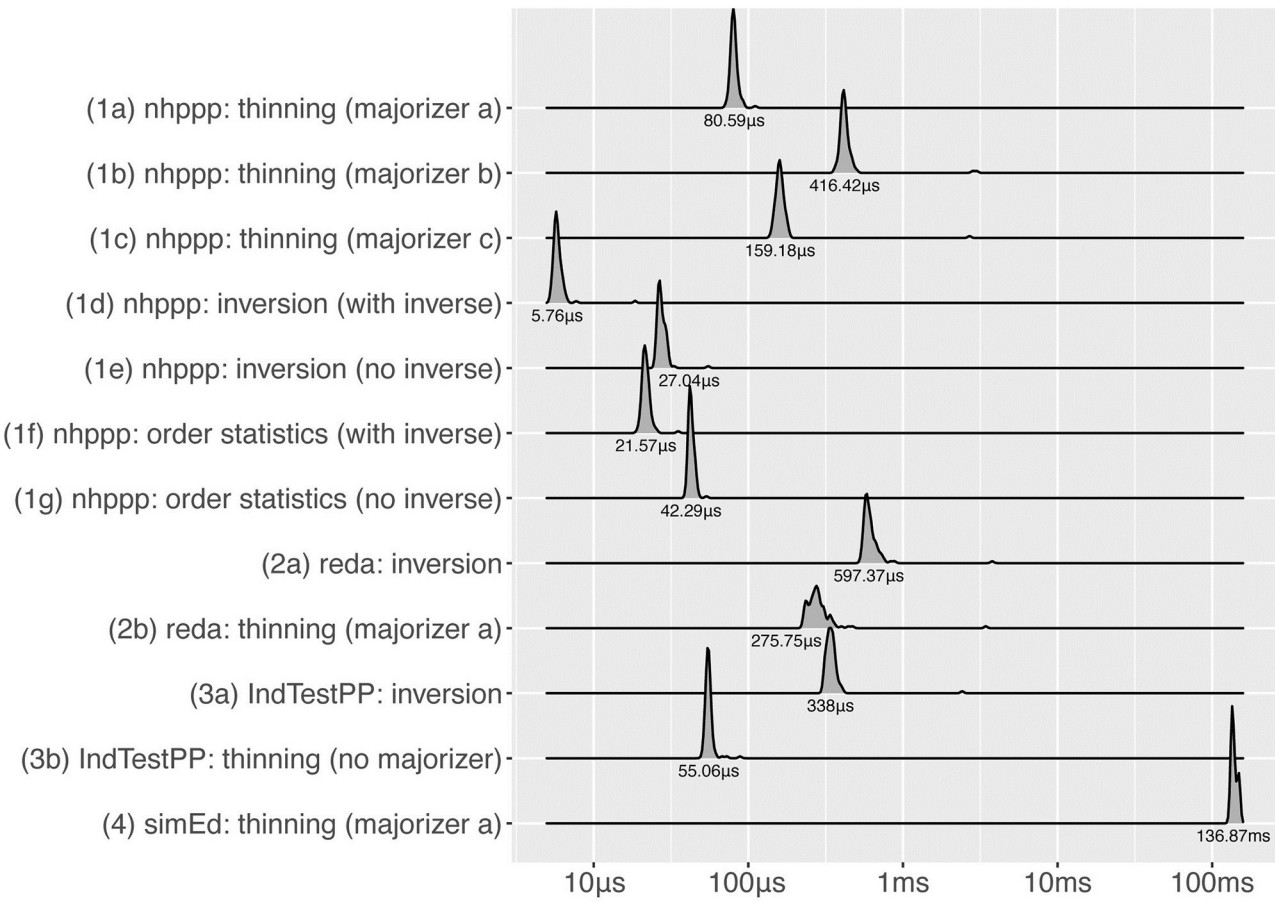

**Fig 7. Computation times when drawing the first event in interval.**

## 9 Computational details and credits

R 4.3.1 [32] was used for all analyses. Packages **xtable** 1.8.4 [33] and **knitr** 1.45 [34] were used for automatic report generation. Packages **ggplot2** 3.4.4 [35], **ggridges** 0.5.5 [36], and **latex2-exp** 0.9.6 [37] were used for plot generation and LaTeXformatting. Packages **nhppp** 0.1.4 [16], **bench** 1.1.3 [38], **rstream** 1.3.7 [39], **otinference** 0.1.0 [40], and **parallel** 4.3.1 were used in the examples and the analyses.

All computations were done on an Apple M1 Max machine with 64 megabytes of random access memory. A preprint of the current paper is in [16]. R itself and all aforementioned packages are available from the Comprehensive R Archive Network (CRAN) at https://CRAN.R-project.org/.

## Supporting information

**S1 Appendix. Piecewise constant majorizer functions.** Algorithm for the automatic generation of piecewise constant majorizer functions.
(PDF)

**S2 Appendix. Conditional sampling from NHPPPs.** Algorithm to sample conditionally on observing at least $m$ events in $(a, b]$.
(PDF)

**S1 Code. Code to reproduce the exhibits.** R code to reproduce the exhibits. Timing results are machine and platform specific.
(TXT)

## Acknowledgments

We thank the investigators of the Cancer Incidence and Surveillance Modeling Network (CIS-NET) Bladder Cancer Site Stavroula Chrysanthopoulou, Jonah Popp, Fernando Alarid-Escudero, Hawre Jalal, and David Garibay for useful discussions.

## Author Contributions

**Conceptualization:** Thomas A. Trikalinos, Yuliia Sereda.

**Formal analysis:** Thomas A. Trikalinos, Yuliia Sereda.

**Funding acquisition:** Thomas A. Trikalinos.

**Investigation:** Thomas A. Trikalinos, Yuliia Sereda.

**Methodology:** Thomas A. Trikalinos, Yuliia Sereda.

**Project administration:** Thomas A. Trikalinos, Yuliia Sereda.

**Software:** Thomas A. Trikalinos, Yuliia Sereda.

**Supervision:** Thomas A. Trikalinos.

**Validation:** Thomas A. Trikalinos.

**Writing – original draft:** Thomas A. Trikalinos, Yuliia Sereda.

**Writing – review & editing:** Thomas A. Trikalinos, Yuliia Sereda.

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
