## [Decision Letter · Decision Letter 0]

22 Apr 2024

PONE-D-24-07749The nhppp package for simulating non-homogeneous Poisson point processes in R.PLOS ONE

Dear Dr. Trikalinos,

Thank you for submitting your manuscript to PLOS ONE. After careful consideration, we feel that it has merit but does not fully meet PLOS ONE’s publication criteria as it currently stands. Therefore, we invite you to submit a revised version of the manuscript that addresses the points raised during the review process.

We look forward to receiving your revised manuscript.

Kind regards,

Mohamed R. Abonazel, Ph.D.

Academic Editor

PLOS ONE

Journal Requirements:

Additional Editor Comments:

1- One of the main problems in regression analysis is Multicollinearity; why not add it to your study? At least add this to future work.

2- Why do you use the “hat” in Equations 1 and 2? The regression model must be without the “hat”. Correct it.

Reviewers' comments:

Reviewer's Responses to Questions

**Comments to the Author**

1. Is the manuscript technically sound, and do the data support the conclusions?

Reviewer #1: Partly

Reviewer #2: Partly

2. Has the statistical analysis been performed appropriately and rigorously? 

Reviewer #1: Yes

Reviewer #2: Yes

3. Have the authors made all data underlying the findings in their manuscript fully available?

Reviewer #1: Yes

Reviewer #2: Yes

4. Is the manuscript presented in an intelligible fashion and written in standard English?

Reviewer #1: Yes

Reviewer #2: No

5. Review Comments to the Author

Reviewer #1: The scholarly paper in question present itself as both methodically well-structured and intellectually robust. From the meticulously curated introduction that sets a well-defined premise, down to the scrupulous alignment of its theoretical framework and literature review.

Reviewer #2: I recommend accepting this paper, but after making the following modifications:

1- I think that the abstract needs improvement.

2- I think that some recent papers related to this research should be mentioned.

3- At the conclusion of this work, the limitations of this research should be mentioned.

6. PLOS authors have the option to publish the peer review history of their article (what does this mean?). If published, this will include your full peer review and any attached files.

Reviewer #1: **Yes: **Suzan Abdel-Rahman

Reviewer #2: No

---

## [Author Response · Author response to Decision Letter 0]

30 May 2024

Please see uploaded pdf "Response_to_comments.pdf".

---

## [Decision Letter · Decision Letter 1]

15 Jul 2024

PONE-D-24-07749R1The nhppp package for simulating non-homogeneous Poisson point processes in R.PLOS ONE

Dear Dr. Trikalinos,

Thank you for submitting your manuscript to PLOS ONE. After careful consideration, we feel that it has merit but does not fully meet PLOS ONE’s publication criteria as it currently stands. Therefore, we invite you to submit a revised version of the manuscript that addresses the points raised during the review process.

We look forward to receiving your revised manuscript.

Kind regards,

Mohamed R. Abonazel, Ph.D.

Academic Editor

PLOS ONE

Journal Requirements:

Reviewers' comments:

Reviewer's Responses to Questions

**Comments to the Author**

1. If the authors have adequately addressed your comments raised in a previous round of review and you feel that this manuscript is now acceptable for publication, you may indicate that here to bypass the “Comments to the Author” section, enter your conflict of interest statement in the “Confidential to Editor” section, and submit your "Accept" recommendation.

Reviewer #2: (No Response)

Reviewer #3: All comments have been addressed

Reviewer #4: (No Response)

Reviewer #5: All comments have been addressed

Reviewer #6: All comments have been addressed

2. Is the manuscript technically sound, and do the data support the conclusions?

Reviewer #2: Partly

Reviewer #3: Yes

Reviewer #4: Partly

Reviewer #5: Yes

Reviewer #6: Yes

3. Has the statistical analysis been performed appropriately and rigorously? 

Reviewer #2: I Don't Know

Reviewer #3: Yes

Reviewer #4: No

Reviewer #5: Yes

Reviewer #6: Yes

4. Have the authors made all data underlying the findings in their manuscript fully available?

Reviewer #2: No

Reviewer #3: Yes

Reviewer #4: Yes

Reviewer #5: Yes

Reviewer #6: Yes

5. Is the manuscript presented in an intelligible fashion and written in standard English?

Reviewer #2: Yes

Reviewer #3: Yes

Reviewer #4: No

Reviewer #5: Yes

Reviewer #6: Yes

6. Review Comments to the Author

Reviewer #2: The authors did not make any of the modifications we requested.

-- -- -- -- --- --- -- -- -- -- --- --- -- -- -- -- --- ---

Reviewer #3: The results of this paper are very useful in studies assuming a NHPP modeling approach with different parametric intensity functions. In many situations, the statistical properties under a classical or a Bayesian approach are verified using simulated data assuming specified parametric intensity functions in NHPP. This point should be discussed in the introduction section. Although, the importance of the study, the review of the literature on NHPP should be improved as motivation for the use of the simulation codes (a software package in R) for different structures of NHPP introduced in the paper.

In my opinion, the paper could be published after improving the motivation for the need of simulated data assuming a NHPP modeling approach and the inclusion of new references on the use of NHPP in the introduction section since the use of NHPP in different applications (environment, epidemiology, economy, engineering) is becoming very popular as observed in the literature. Usually a NHPP modeling approach is considered assuming different parametric intensity functions, sometimes in presence of one or more change-points, under a Bayesian approach given the difficulties to obtain accurate classical inferences (maximum likelihood estimates and related usual asymptotical results). In summary: it is needed to introduce a complete review of the literature on applications of NHPP (new references on the use of NHPP in different areas) in the introduction section to have better motivation for the study.

Reviewer #4: The proposed manuscript is good and has some merit, but it is not adequate for the PLOS journal. The authors should send it to R Journal, since it shows the performance of a R package.

Reviewer #5: This manuscript presents the nhppp package for simulating events from one-dimensional non-homogeneous Poisson point processes (NHPPPs) in R. Related work has already been published as a preprint. While the manuscript is comprehensive and detailed, it currently resembles a user manual for the toolbox. Before considering acceptance, I recommend that the authors supplement the conclusion with quantitative results, particularly highlighting the advantages of their package over existing toolboxes, such as improvements in efficiency and accuracy, among other metrics.

Reviewer #6: This research is written and formatted in a respectable academic style comparable to studies published in the R journal. While I recommend that authors try to publish it in the R journal, because it is the best place for this type of research, PLOS ONE also accepts these types of studies. Therefore, I recommend accepting publication in PLOS ONE after reviewing and ensuring that it conforms to the journal's requirements.

7. PLOS authors have the option to publish the peer review history of their article (what does this mean?). If published, this will include your full peer review and any attached files.

Reviewer #2: No

Reviewer #3: No

Reviewer #4: No

Reviewer #5: No

Reviewer #6: No

---

## [Author Response · Author response to Decision Letter 1]

15 Jul 2024

Please see the uploaded pdf "response_to_reviewers.pdf"

---

## [Decision Letter · Decision Letter 2]

9 Sep 2024

The nhppp package for simulating non-homogeneous Poisson point processes in R.

PONE-D-24-07749R2

Dear Dr. Trikalinos,

We’re pleased to inform you that your manuscript has been judged scientifically suitable for publication and will be formally accepted for publication once it meets all outstanding technical requirements.

Kind regards,

Abhik Ghosh

Academic Editor

PLOS ONE

Additional Editor Comments (optional):

Reviewers' comments:

Reviewer's Responses to Questions

**Comments to the Author**

1. If the authors have adequately addressed your comments raised in a previous round of review and you feel that this manuscript is now acceptable for publication, you may indicate that here to bypass the “Comments to the Author” section, enter your conflict of interest statement in the “Confidential to Editor” section, and submit your "Accept" recommendation.

Reviewer #3: All comments have been addressed

2. Is the manuscript technically sound, and do the data support the conclusions?

Reviewer #3: Yes

3. Has the statistical analysis been performed appropriately and rigorously? 

Reviewer #3: Yes

4. Have the authors made all data underlying the findings in their manuscript fully available?

Reviewer #3: Yes

5. Is the manuscript presented in an intelligible fashion and written in standard English?

Reviewer #3: Yes

6. Review Comments to the Author

Reviewer #3: The paper is now suitable for publication.This paper is useful in applications of NHPP(computer program in R).

7. PLOS authors have the option to publish the peer review history of their article (what does this mean?). If published, this will include your full peer review and any attached files.

Reviewer #3: No

---

## [Editor Report · Acceptance letter]

19 Sep 2024

PONE-D-24-07749R2 

PLOS ONE

Dear Dr. Trikalinos, 

I'm pleased to inform you that your manuscript has been deemed suitable for publication in PLOS ONE. Congratulations! Your manuscript is now being handed over to our production team.

Kind regards, 

on behalf of

Dr. Abhik Ghosh 

Academic Editor

PLOS ONE